# Computing the Stationary Distribution, *Locally*

**Christina E. Lee**
LIDS, Department of EECS
Massachusetts Institute of Technology
celee@mit.edu

**Asuman Ozdaglar**
LIDS, Department of EECS
Massachusetts Institute of Technology
asuman@mit.edu

**Devavrat Shah**
Department of EECS
Massachusetts Institute of Technology
devavrat@mit.edu

## Abstract

Computing the stationary distribution of a large finite or countably infinite state space Markov Chain (MC) has become central in many problems such as statistical inference and network analysis. Standard methods involve large matrix multiplications as in power iteration, or simulations of long random walks, as in Markov Chain Monte Carlo (MCMC). Power iteration is costly, as it involves computation at every state. For MCMC, it is difficult to determine whether the random walks are long enough to guarantee convergence. In this paper, we provide a novel algorithm that answers whether a chosen state in a MC has stationary probability larger than some $\Delta \in (0, 1)$, and outputs an estimate of the stationary probability. Our algorithm is constant time, using information from a *local neighborhood* of the state on the graph induced by the MC, which has constant size relative to the state space. The multiplicative error of the estimate is upper bounded by a function of the mixing properties of the MC. Simulation results show MCs for which this method gives tight estimates.

## 1 Introduction

Computing the stationary distribution of a Markov chain (MC) with a very large state space (finite, or countably infinite) has become central to statistical inference. The ability to tractably simulate MCs along with the generic applicability has made Markov Chain Monte Carlo (MCMC) a method of choice and arguably the top algorithm of the twentieth century [1]. However, MCMC and its variations suffer from limitations in large state spaces, motivating the development of super-computation capabilities – be it nuclear physics [2, Chapter 8], Google's computation of PageRank [3], or stochastic simulation at-large [4]. MCMC methods involve sampling states from a long random walk over the entire state space [5, 6]. It is difficult to determine when the algorithm has walked "long enough" to produce reasonable approximations for the stationary distribution.

Power iteration is another method commonly used for computing leading eigenvectors and stationary distributions of MCs. The method involves iterative multiplication of the transition matrix of the MC [7]. However, there is no clearly defined stopping condition in general settings, and computations must be performed at every state of the MC.

In this paper, we provide a novel algorithm that addresses these limitations. Our algorithm answers the following question: for a given node $i$ of a countable state space MC, is the stationary probability of $i$ larger a given threshold $\Delta \in (0, 1)$, and can we approximate it? For chosen parameters $\Delta, \epsilon$, and $\alpha$, our algorithm guarantees that for nodes such that the estimate $\hat{\pi}_i < \Delta/(1 + \epsilon)$, the true

value $\pi_i$ is also less than $\Delta$ with probability at least $1 - \alpha$. In addition, if $\hat{\pi}_i \geq \Delta/(1 + \epsilon)$, with probability at least $1 - \alpha$, the estimate is within an $\epsilon$ times $Z_{\max}(i)$ multiplicative factor away from the true $\pi_i$, where $Z_{\max}(i)$ is effectively a "local mixing time" for $i$ derived from the fundamental matrix of the transition probability matrix $P$.

The running time of the algorithm is upper bounded by $\tilde{O}\left(\frac{\ln(\frac{1}{\alpha})}{\epsilon^3 \Delta}\right)$, which is constant with respect to the MC. Our algorithm uses only a"local" neighborhood of the state $i$, defined with respect to the Markov graph. Stopping conditions are easy to verify and have provable performance guarantees. Its correctness relies on a basic property: the stationary probability of each node is inversely proportional to the mean of its "return time." Therefore, we sample return times to the node and use the empirical average as an estimate. Since return times can be arbitrarily long, we truncate sample return times at a chosen threshold. Hence, our algorithm is a *truncated Monte Carlo* method.

We utilize the exponential concentration of return times in Markov chains to establish theoretical guarantees for the algorithm. For finite state Markov chains, we use results from Aldous and Fill [8]. For countably infinite state space Markov chains, we build upon a result by Hajek [9] on the concentration of certain types of hitting times to derive concentration of return times to a given node. We use these concentration results to upper bound the estimation error and the algorithm runtime as a function of the truncation threshold and the mixing properties of the graph. For graphs that mix quickly, the distribution over return times concentrates more sharply around its mean, resulting in tighter performance guarantees. We illustrate the wide applicability of our local algorithm for computing network centralities and stationary distributions of queuing models.

**Related Work.** MCMC was originally proposed in [5], and a tractable way to design a random walk for a target distribution was proposed by Hastings [6]. Given a distribution $\pi(x)$, the method designs a Markov chain such that the stationary distribution of the Markov chain is equal to the target distribution. Without using the full transition matrix of the Markov chain, Monte Carlo sampling techniques estimate the distribution by sampling random walks via the transition probabilities at each node. As the length of the random walk approaches infinity, the distribution over possible states of the random walk approaches stationary distribution. Articles by Diaconis and Saloff-Coste [10] and Diaconis [11] provide a summary of major developments from a probability theoretic perspective.

The majority of work following the initial introduction of the algorithm involves analyzing the convergence rates and mixing times of this random walk [8, 12]. Techniques involve spectral analysis or coupling arguments. Graph properties such as conductance help characterize the graph spectrum for reversible Markov chains. For general non-reversible countably infinite state space Markov chains, little is known about the mixing time. Thus, it is difficult to verify if the random walk has sufficiently converged to the stationary distribution, and before that point there is no guarantee whether the estimate obtained from the random walk is larger or smaller than the true stationary probability.

Power iteration is an equally old and well-established method for computing leading eigenvectors of matrices [7]. Given a matrix $A$ and a seed vector $x_0$, power iteration recursively computes $x_{t+1} = \frac{Ax_t}{\|Ax_t\|}$. The convergence rate of $x_t$ to the leading eigenvector is governed by the spectral gap. As mentioned above, techniques for analyzing the spectrum are not well developed for general non-reversible MCs, thus it is difficult to know how many iterations are sufficient. Although power iteration can be implemented in a distributed manner, each iteration requires computation to be performed by every state in the MC, which is expensive for large state space MCs. For countably infinite state space MCs, there is no clear analog to matrix multiplication.

In the specialized setting of PageRank, the goal is to compute the stationary distribution of a specific Markov chain described by a transition matrix $P = (1 - \beta)Q + \beta \mathbf{1} \cdot r^T$, where $Q$ is a stochastic transition probability matrix, and $\beta$ is a scalar in $(0, 1)$. This can be interpreted as random walk in which every step either follows $Q$ with probability $1 - \beta$, or with probability $\beta$ jumps to a node according to the distribution specified by vector $r$. By exploiting this special structure, numerous recent results have provided local algorithms for computing PageRank efficiently. This includes work by Jeh and Widom [13], Fogaras *et al.* [14], Avrachenkov *et al.* [15], Bahmani *et al.* [16] and most recently, Borgs *et al.* [17]: it outputs a set of "important" nodes – with probability $1 - o(1)$, it includes all nodes with PageRank greater than a given threshold $\Delta$, and does not include nodes with PageRank less than $\Delta/c$ for a given $c > 1$. The algorithm runs in time $O\left(\frac{1}{\Delta} \text{polylog}(n)\right)$. Unfortunately, these approaches are specific to PageRank and do not extend to general MCs.

## 2 Setup, problem statement & algorithm

Consider a discrete time, irreducible, positive-recurrent MC $\{X_t\}_{t\geq 0}$ on a countable state space $\Sigma$ having transition probability matrix $P$. Let $P_{ij}^{(n)}$ be the $(i,j)$-coordinate of $P^n$ such that

$$P_{ij}^{(n)} \triangleq \mathbb{P}(X_n = j | X_0 = i).$$

Throughout the paper, we will use the notation $\mathbb{E}_i[\cdot] = \mathbb{E}[\cdot | X_0 = i]$, and $\mathbb{P}_i(\cdot) = \mathbb{P}(\cdot | X_0 = i)$. Let $T_i$ be the *return time* to a node $i$, and let $H_i$ be the *maximal hitting time* to a node $i$ such that

$$T_i = \inf\{t \geq 1 \mid X_t = i\} \quad \text{and} \quad H_i = \max_{j \in \Sigma} \mathbb{E}_j[T_i]. \tag{1}$$

The *stationary distribution* is a function $\pi : \Sigma \rightarrow [0,1]$ such that $\sum_{i \in \Sigma} \pi_i = 1$ and $\pi_i = \sum_{j \in \Sigma} \pi_j P_{ji}$ for all $i \in \Sigma$. An irreducible positive recurrent Markov chain has a unique stationary distribution satisfying [18, 8]:

$$\pi_i = \frac{\mathbb{E}_i \left[ \sum_{t=1}^{T_i} \mathbf{1}_{\{X_t = i\}} \right]}{\mathbb{E}_i[T_i]} = \frac{1}{\mathbb{E}_i[T_i]} \quad \text{for all} \quad i \in \Sigma. \tag{2}$$

The Markov chain can be visualized as a random walk over a weighted directed graph $G = (\Sigma, E, P)$, where $\Sigma$ is the set of nodes, $E = \{(i,j) \in \Sigma \times \Sigma : P_{ij} > 0\}$ is the set of edges, and $P$ describes the weights of the edges.[1] The *local neighborhood of size $r$* around node $i \in \Sigma$ is defined as $\{j \in \Sigma \mid d_G(i,j) \leq r\}$, where $d_G(i,j)$ is the length of the shortest directed path (in terms of number of edges) from $i$ to $j$ in $G$. An algorithm is *local* if it only uses information within a local neighborhood of size $r$ around $i$, where $r$ is constant with respect to the size of the state space.

The *fundamental matrix $Z$* of a finite state space Markov chain is

$$Z \triangleq \sum_{t=0}^{\infty} \left( P^{(t)} - \mathbf{1}\pi^T \right) = \left( I - P + \mathbf{1}\pi^T \right)^{-1}, \text{ such that } Z_{jk} \triangleq \sum_{t=0}^{\infty} \left( P_{jk}^{(t)} - \pi_k \right).$$

Since $P_{jk}^{(t)}$ denotes the probability that a random walk beginning at node $j$ is at node $k$ after $t$ steps, $Z_{jk}$ represents how quickly the probability mass at node $k$ from a random walk beginning at node $j$ converges to $\pi_k$. We will use this to provide bounds for the performance of our algorithm.

### 2.1 Problem Statement

Consider a discrete time, irreducible, aperiodic, positive recurrent MC $\{X_t\}_{t\geq 0}$ on a countable state space $\Sigma$ with transition probability matrix $P : \Sigma \times \Sigma \rightarrow [0,1]$. Given node $i$ and threshold $\Delta$, is $\pi_i > \Delta$? If so, what is $\pi_i$? We answer this with a local algorithm, which uses only edges within a local neighborhood around $i$ of constant size with respect to the state space.

We illustrate the limitations of using a local algorithm for answering this question. Consider the Clique-Cycle Markov chain shown in Figure 1(a) with $n$ nodes, composed of a size $k$ clique connected to a size $(n - k + 1)$ cycle. For node $j$ in the clique excluding $i$, with probability $1/2$, the random walk stays at node $j$, and with probability $1/2$ the random walk chooses a random neighbor uniformly. For node $j$ in the cycle, with probability $1/2$, the random walk stays at node $j$, and with probability $1/2$ the random walk travels counterclockwise to the subsequent node in the cycle. For node $i$, with probability $\epsilon$ the random walk enters the cycle, with probability $1/2$ the random walk chooses any neighbor in the clique; and with probability $1/2 - \epsilon$ the random walk stays at node $i$. We can show that the expected return time to node $i$ is $(1 - 2\epsilon)k + 2\epsilon n$.

Therefore, $\mathbb{E}_i[T_i]$ scales linearly in $n$ and $k$. Suppose we observe only the local neighborhood of constant size $r$ around node $i$. All Clique-Cycle Markov chains with more than $k + 2r$ nodes have identical local neighborhoods. Therefore, for any $\Delta \in (0,1)$, there exists two Clique-Cycle Markov chains which have the same $\epsilon$ and $k$, but two different values for $n$, such that even though their local neighborhoods are identical, $\pi_i > \Delta$ in the MC with a smaller $n$, while $\pi_i < \Delta$ in the MC with a larger $n$. Therefore, by restricting ourselves to a local neighborhood around $i$ of constant size, we will not be able to correctly determine whether $\pi_i > \Delta$ for every node $i$ in any arbitrary MC.

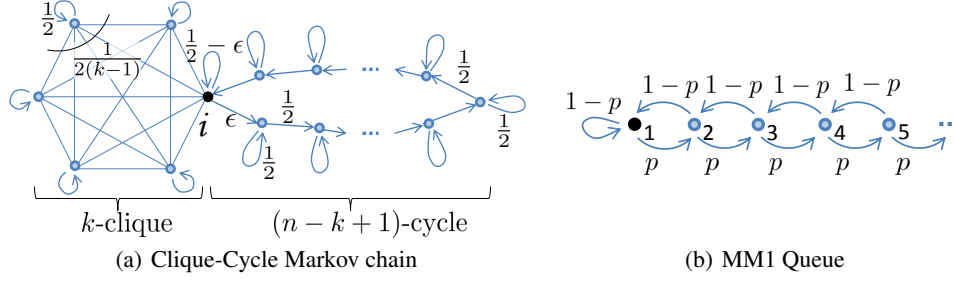

(a) Clique-Cycle Markov chain      (b) MM1 Queue

Figure 1: Examples of Markov Chains

## 2.2 Algorithm

Given a threshold $\Delta \in (0,1)$ and a node $i \in \Sigma$, the algorithm obtains an estimate $\hat{\pi}_i$ of $\pi_i$, and uses $\hat{\pi}_i$ to determine whether to output 0 ($\pi_i \leq \Delta$) or 1 ($\pi_i > \Delta$). The algorithm relies on the characterization of $\pi_i$ given in Eq. (2): $\pi_i = 1/\mathbb{E}_i[T_i]$. It takes many independent samples of a truncated random walk that begins at node $i$ and stops either when the random walk returns to node $i$, or when the length exceeds a predetermined maximum denoted by $\theta$. Each sample is generated by simulating the random walk using "crawl" operations over the MC. The expected length of each random walk sample is $\mathbb{E}_i[\min(T_i, \theta)]$, which is close to $\mathbb{E}_i[T_i]$ when $\theta$ is large.

As the number of samples and $\theta$ go to infinity, the estimate will converge almost surely to $\pi_i$, due to the strong law of large numbers and positive recurrence of the MC. We use Chernoff's bound to choose a sufficiently large number of samples as a function of $\theta$ to guarantee that with probability $1 - \alpha$, the average length of the sample random walks will lie within $(1 \pm \epsilon)$ of $\mathbb{E}_i[\min(T_i, \theta)]$.

We also need to choose an suitable value for $\theta$ that balances between accuracy and computation cost. The algorithm searches for an appropriate size for the local neighborhood by beginning small and increasing the size geometrically. In our analysis, we will show that the total computation summed over all iterations is only a constant factor more than the computation in the final iteration.

---

**Input:** Anchor node $i \in \Sigma$ and parameters $\Delta$ = threshold for importance, $\epsilon$ = closeness of the estimate, and $\alpha$ = probability of failure.

**Initialize:** Set
$$t = 1, \theta^{(1)} = 2, N^{(1)} = \left\lceil \frac{6(1+\epsilon)\ln(8/\alpha)}{\epsilon^2} \right\rceil.$$

**Step 1 (Gather Samples)** For each $k$ in $\{1, 2, 3, \ldots, N^{(t)}\}$, generate independent samples $s_k \sim \min(T_i, \theta^{(t)})$ by simulating paths of the MC beginning at node $i$, and setting $s_k$ to be the length of the $k^{th}$ sample path. Let $\hat{p}^{(t)}$ = fraction of samples truncated at $\theta^{(t)}$,

$$\hat{T}_i^{(t)} = \frac{1}{N^{(t)}} \sum_{k=1}^{N^{(t)}} s_k, \quad \hat{\pi}_i^{(t)} = \frac{1}{\hat{T}_i^{(t)}}, \quad \text{and} \quad \tilde{\pi}_i^{(t)} = \frac{1 - \hat{p}^{(t)}}{\hat{T}_i^{(t)}}.$$

**Step 2 (Termination Conditions)**

- If **(a)** $\hat{\pi}_i^{(t)} < \frac{\Delta}{(1+\epsilon)}$, then stop and return **0**, and estimates $\hat{\pi}_i^{(t)}$ and $\tilde{\pi}_i^{(t)}$.

- Else if **(b)** $\hat{p}^{(t)} \cdot \hat{\pi}_i^{(t)} < \epsilon\Delta$, then stop and return **1**, and estimates $\hat{\pi}_i^{(t)}$ and $\tilde{\pi}_i^{(t)}$.

- Else continue.

**Step 3 (Update Rules)** Set
$$\theta^{(t+1)} \leftarrow 2 \cdot \theta^{(t)}, N^{(t+1)} \leftarrow \left\lceil \frac{3(1+\epsilon)\theta^{(t+1)}\ln(4\theta^{(t+1)}/\alpha)}{\hat{T}_i^{(t)}\epsilon^2} \right\rceil, \text{ and } t \leftarrow t + 1.$$

Return to **Step 1**.

**Output:** **0** or **1** indicating whether $\pi_i > \Delta$, and estimates $\hat{\pi}_i^{(t)}$ and $\tilde{\pi}_i^{(t)}$.

---

This algorithm outputs two estimates for the anchor node $i$: $\hat{\pi}_i$, which relies on the second expression in Eq. (2), and $\tilde{\pi}_i$, which relies on the first expression in Eq. (2). We refer to the total number of iterations used in the algorithm as the value of $t$ at the time of termination, denoted by $t_{\max}$. The total number of random walk steps taken within the first $t$ iterations is $\sum_{k=1}^{t} N^{(t)} \cdot \hat{T}_i^{(t)}$.

The algorithm will always terminate within $\ln\left(\frac{1}{\epsilon\Delta}\right)$ iterations. Since $\theta^{(t)}$ governs the radius of the local neighborhood that the algorithm utilizes, this implies that our algorithm is local, since the maximum distance is strictly upper bounded by $\frac{1}{\epsilon\Delta}$, which is constant with respect to the MC.

With high probability, the estimate $\hat{\pi}_i^{(t)}$ is larger than $\frac{\pi_i}{1+\epsilon}$ due to the truncation. Thus when the algorithm terminates at stopping condition (a), $\pi_i < \Delta$ with high probability. When the algorithm terminates at condition (b), the fraction of samples truncated is small, which will imply that the percentage error of estimate $\hat{\pi}_i^{(t)}$ is upper bounded as a function of $\epsilon$ and properties of the MC.

## 3  Theoretical guarantees

The following theorems give correctness and convergence guarantees for the algorithm. The proofs have been omitted and can be found in the extended version of this paper [19].

**Theorem 3.1.** *For an aperiodic, irreducible, positive recurrent, countable state space Markov chain, and for any $i \in \Sigma$, with probability greater than $1 - \alpha$:*

1. **Correctness.** *For all iterations $t$, $\hat{\pi}_i^{(t)} \geq \frac{\pi_i}{1+\epsilon}$. Therefore, if the algorithm terminates at condition (a) and outputs 0, then $\pi_i < \Delta$.*

2. **Convergence.** *The number of iterations $t_{\max}$ and the total number of steps (or neighbor queries) used by the algorithm are bounded above by[2][3]*

$$t_{\max} \leq \ln\left(\frac{1}{\epsilon\Delta}\right), \quad and \quad \sum_{k=1}^{t_{\max}} N^{(t)} \cdot \hat{T}_i^{(t)} \leq \tilde{O}\left(\frac{\ln(\frac{1}{\alpha})}{\epsilon^3\Delta}\right).$$

Part 1 is proved by using Chernoff's bound to show that $N^{(t)}$ is large enough to guarantee that with probability greater than $1 - \alpha$, for all iterations $t$, $\hat{T}_i^{(t)}$ concentrates around its mean. Part 2 asserts that the algorithm terminates in finite time as a function of the parameters of the algorithm, independent from the size of the MC state space. Therefore this implies that our algorithm is local. This theorem holds for all aperiodic, irreducible, positive recurrent MCs. This is proved by observing that $\hat{T}_i^{(t)} > \hat{p}^{(t)}\theta^{(t)}$. Therefore when $\theta^{(t)} > \frac{1}{\epsilon\Delta}$, termination condition (b) must be satisfied.

### 3.1  Finite-state space Markov Chain

We can obtain characterizations for the approximation error and the running time as functions of specific properties of the MC. The analysis depends on how sharply the distribution over return times concentrates around the mean.

**Theorem 3.2.** *For an irreducible Markov chain $\{X_t\}$ with finite state space $\Sigma$ and transition probability matrix $P$, for any $i \in \Sigma$, with probability greater than $1 - \alpha$, for all iterations $t$,*

$$\left|\frac{\hat{\pi}_i^{(t)} - \pi_i}{\hat{\pi}_i^{(t)}}\right| \leq 2(1-\epsilon)\mathbb{P}_i(T_i > \theta^{(t)})Z_{\max}(i) + \epsilon \leq 4(1-\epsilon)2^{-\theta^{(t)}/2H_i}Z_{\max}(i) + \epsilon,$$

*where $H_i$ is defined in Eq (1), and $Z_{max}(i) = \max_j |Z_{ji}|$.*

*Therefore, with probability greater than $1 - \alpha$, if the algorithm terminates at condition (b), then*

$$\left|\frac{\hat{\pi}_i^{(t)} - \pi_i}{\hat{\pi}_i^{(t)}}\right| \leq \epsilon\left(3Z_{\max}(i) + 1\right).$$

Theorem 3.2 shows that the percentage error in the estimate $\hat{\pi}_i^{(t)}$ decays exponentially in $\theta^{(t)}$, which doubles in each iteration. The proof relies on the fact that the distribution of the return time $T_i$ has an exponentially decaying tail [8], ensuring that the return time $T_i$ concentrates around its mean $\mathbb{E}_i[T_i]$. When the algorithm terminates at stopping condition (b), $\mathbb{P}(T_i > \theta) \leq \epsilon(\frac{4}{3} + \epsilon)$ with high probability, thus the percentage error is bounded by $O(\epsilon Z_{\max}(i))$.

Similarly, we can analyze the error between the second estimate $\tilde{\pi}_i^{(t)}$ and $\pi_i$, in the case when $\theta^{(t)}$ is large enough such that $\mathbb{P}(T_i > \theta^{(t)}) < \frac{1}{2}$. This is required to guarantee that $(1 - \hat{p}^{(t)})$ lies within an $\epsilon$ multiplicative interval around its mean with high probability. Observe than $2Z_{\max}(i)$ is replaced by $\max(2Z_{\max}(i) - 1, 1)$. Thus for some values of $Z_{\max}(i)$, the error bound for $\tilde{\pi}_i$ is smaller than the equivalent bound for $\hat{\pi}_i$. We will show simulations of computing PageRank, in which $\tilde{\pi}_i$ estimates $\pi_i$ more closely than $\hat{\pi}_i$.

**Theorem 3.3.** *For an irreducible Markov chain $\{X_t\}$ with finite state space $\Sigma$ and transition probability matrix $P$, for any $i \in \Sigma$, with probability greater than $1 - \alpha$, for all iterations $t$ such that $\mathbb{P}(T_i > \theta^{(t)}) < \frac{1}{2}$,*

$$\left| \frac{\tilde{\pi}_i^{(t)} - \pi_i}{\tilde{\pi}_i^{(t)}} \right| \leq \left( \frac{1+\epsilon}{1-\epsilon} \right) \left( \frac{\mathbb{P}_i(T_i > \theta^{(t)})}{1 - \mathbb{P}_i(T_i > \theta^{(t)})} \right) \max(2Z_{\max}(i) - 1, 1) + \frac{2\epsilon}{1-\epsilon}.$$

Theorem 3.4 also uses the property of an exponentially decaying tail as a function of $H_i$ to show that for large $\theta^{(t)}$, with high probability, $\mathbb{P}_i\left(T_i > \theta^{(t)}\right)$ will be small and $\hat{\pi}_i^{(t)}$ will be close to $\pi_i$, and thus the algorithm will terminate at one of the stopping conditions. The bound is a function of how sharply the distribution over return times concentrates around the mean. Theorem 3.4(a) states that for low probability nodes, the algorithm will terminate at stopping condition (a) for large enough iterations. Theorem 3.4(b) states that for all nodes, the algorithm will terminate at stopping condition (b) for large enough iterations.

**Theorem 3.4.** *For an irreducible Markov chain $\{X_t\}$ with finite state space $\Sigma$,*
**(a)** *For any node $i \in \Sigma$ such that $\pi_i < (1 - \epsilon)\Delta/(1 + \epsilon)$, with probability greater than $1 - \alpha$, the total number of steps used by the algorithm is bounded above by*

$$\sum_{k=1}^{t_{\max}} N^{(t)} \cdot \hat{T}_i^{(t)} \leq \tilde{O}\left( \frac{\ln(\frac{1}{\alpha})}{\epsilon^2} \left( H_i \ln \left( \left( \frac{1}{1 - 2^{-1/2H_i}} \right) \left( \frac{1}{\pi_i} - \frac{1+\epsilon}{(1-\epsilon)\Delta} \right)^{-1} \right) \right) \right).$$

**(b)** *For all nodes $i \in \Sigma$, with probability greater than $1 - \alpha$, the total number of steps used by the algorithm is bounded above by*

$$\sum_{k=1}^{t_{\max}} N^{(t)} \cdot \hat{T}_i^{(t)} \leq \tilde{O}\left( \frac{\ln(\frac{1}{\alpha})}{\epsilon^2} \left( \frac{H_i}{\alpha} \ln \left( \pi_i \left( \frac{1}{\epsilon\Delta} + \frac{1}{1 - 2^{-1/2H_i}} \right) \right) \right) \right).$$

### 3.2 Countable-state space Markov Chain

The proofs of Theorems 3.2 and 3.4 require the state space of the MC to be finite, so we can upper bound the tail of the distribution of $T_i$ using the maximal hitting time $H_i$. In fact, these results can be extended to many countably infinite state space Markov chains, as well. We prove that the tail of the distribution of $T_i$ decays exponentially for any node $i$ in any countable state space Markov chain that satisfies Assumption 3.5.

**Assumption 3.5.** *The Markov chain $\{X_t\}$ is aperiodic and irreducible. There exists a Lyapunov function $V : \Sigma \to \mathbb{R}_+$ and constants $\nu_{\max}, \gamma > 0$, and $b \geq 0$, that satisfy the following conditions:*

1. *The set $B = \{x \in \Sigma : V(x) \leq b\}$ is finite,*

2. *For all $x, y \in \Sigma$ such that $\mathbb{P}\left(X_{t+1} = j | X_t = i\right) > 0$, $|V(j) - V(i)| \leq \nu_{\max}$,*

3. *For all $x \in \Sigma$ such that $V(x) > b$, $\mathbb{E}\left[V(X_{t+1}) - V(X_t)|X_t = x\right] < -\gamma$.*

At first glance, this assumption may seem very restrictive. But in fact, this is quite reasonable: by the Foster-Lyapunov criteria [20], a countable state space Markov chain is positive recurrent if and

only if there exists a Lyapunov function $V : \Sigma \to \mathbb{R}_+$ that satisfies condition (1) and (3), as well as (2'): $\mathbb{E}[V(X_{t+1})|X_t = x] < \infty$ for all $x \in \Sigma$. Assumption 3.5 has (2), which is a restriction of the condition (2'). The existence of the Lyapunov function allows us to decompose the state space into sets $B$ and $B^c$ such that for all nodes $x \in B^c$, there is an expected decrease in the Lyapunov function in the next step or transition. Therefore, for all nodes in $B^c$, there is a negative drift towards set $B$. In addition, in any single step, the random walk cannot escape "too far".

Using the concentration bounds for the countable state space settings, we can prove the following theorems that parallel the theorems stated for the finite state space setting. The formal statements are restricted to nodes in $B = \{i \in \Sigma : V(i) \le b\}$. This is not actually restrictive, as for any $i$ such that $V(i) > b$, we can define a new Lyapunov function where $V'(i) = b$, and $V'(j) = V(j)$ for all $j \ne i$. Then $B' = B \cup \{i\}$, and $V'$ still satisfies assumption 3.5 for new values of $\nu_{\max}$, $\gamma$, and $b$.

**Theorem 3.6.** *For a Markov chain satisfying Assumption 3.5, for any $i \in B$, with probability greater than $1 - \alpha$, for all iterations $t$,*

$$\left| \frac{\hat{\pi}_i^{(t)} - \pi_i}{\hat{\pi}_i^{(t)}} \right| \le 4(1 - \epsilon) \left( \frac{2^{-\theta^{(t)}/R_i}}{1 - 2^{-1/R_i}} \right) \pi_i + \epsilon,$$

*where $R_i$ is defined such that*

$$R_i = O\left( \frac{H_i^B e^{2\eta\nu_{\max}}}{(1 - \rho)(e^{\eta\nu_{\max}} - \rho)} \right),$$

*and $H_i^B$ is the maximal hitting time over the Markov chain with its state space restricted to the subset $B$. The scalars $\eta$ and $\rho$ are functions of $\gamma$ and $\nu_{\max}$ (defined in [9]).*

**Theorem 3.7.** *For a Markov chain satisfying Assumption 3.5,*
**(a)** *For any node $i \in B$ such that $\pi_i < (1 - \epsilon)\Delta/(1 + \epsilon)$, with probability greater than $1 - \alpha$, the total number of steps used by the algorithm is bounded above by*

$$\sum_{k=1}^{t_{\max}} N^{(t)} \cdot \hat{T}_i^{(t)} \tilde{O}\left( \frac{\ln(\frac{1}{\alpha})}{\epsilon^2} \left( R_i \ln\left( \left( \frac{1}{1 - 2^{-1/R_i}} \right) \left( \frac{1}{\pi_i} - \frac{1 + \epsilon}{(1 - \epsilon)\Delta} \right)^{-1} \right) \right) \right).$$

**(b)** *For all nodes $i \in B$, with probability greater than $1 - \alpha$, the total number of steps used by the algorithm is bounded above by*

$$\sum_{k=1}^{t_{\max}} N^{(t)} \cdot \hat{T}_i^{(t)} \le \tilde{O}\left( \frac{\ln(\frac{1}{\alpha})}{\epsilon^2} \left( \frac{R_i}{\alpha} \ln\left( \pi_i \left( \frac{1}{\epsilon\Delta} + \frac{1}{1 - 2^{-1/R_i}} \right) \right) \right) \right).$$

In order to prove these theorems, we build upon results of [9], and establish that return times have exponentially decaying tails for countable state space MCs that satisfy Assumption 3.5.

## 4 Example applications: PageRank and MM1 Queue

PageRank is frequently used to compute the importance of web pages in the web graph. Given a scalar parameter $\beta$ and a stochastic transition matrix $P$, let $\{X_t\}$ be the Markov chain with transition matrix $\frac{\beta}{n}\mathbf{1} \cdot \mathbf{1}^T + (1 - \beta)P$. In every step, there is an $\beta$ probability of jumping uniformly randomly to any other node in the network. PageRank is defined as the stationary distribution of this Markov chain. We apply our algorithm to compute PageRank on a random graph generated according to the configuration model with a power law degree distribution for $\beta = 0.15$.

In queuing theory, Markov chains are used to model the queue length at a server, which evolves over time as requests arrive and are processed. We use the basic MM1 queue, equivalent to a random walk on $\mathbb{Z}_+$. Assume we have a single server where the requests arrive according to a Poisson process, and the processing time for a single request is distributed exponentially. The queue length is modeled with the Markov chain shown in Figure 1(b), where $p$ is the probability that a new request arrives before the current request is fully processed.

Figures 2(a) and 2(b) plot $\hat{\pi}_i^{(t_{\max})}$ and $\tilde{\pi}_i^{(t_{\max})}$ for each node in the PageRank or MM1 queue MC, respectively. For both examples, we choose algorithm parameters $\Delta = 0.02$, $\epsilon = 0.15$, and $\alpha = 0.2$.

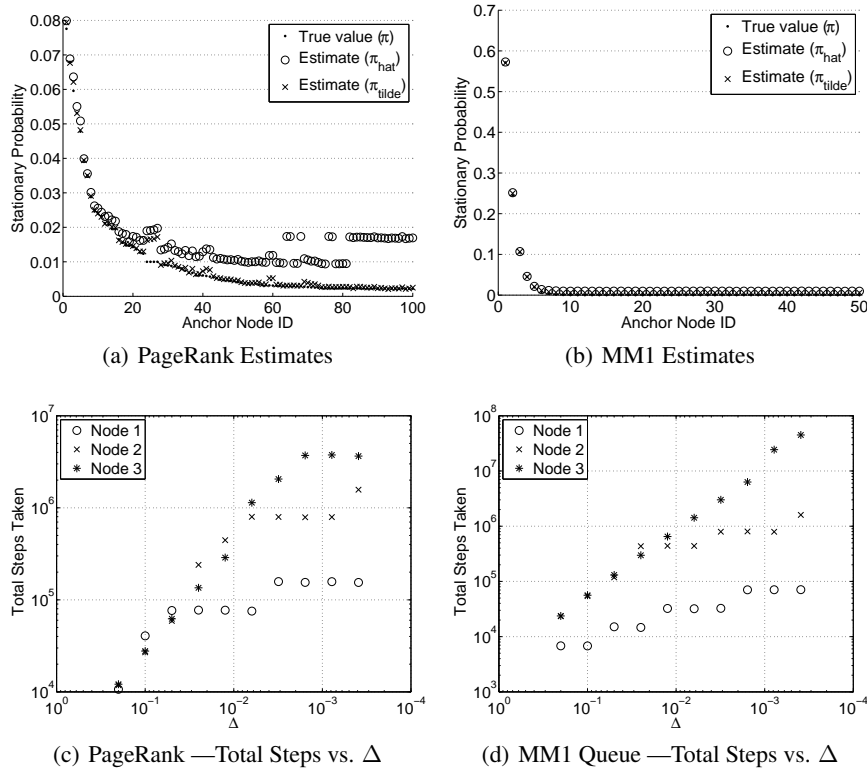

(a) PageRank Estimates

(b) MM1 Estimates

(c) PageRank —Total Steps vs. $\Delta$

(d) MM1 Queue —Total Steps vs. $\Delta$

Figure 2: Simulations showing results of our algorithm applied to PageRank and MM1 Queue setting

Observe that the algorithm indeed obtains close estimates for nodes such that $\pi_i > \Delta$, and for nodes such that $\pi_i \leq \Delta$, the algorithm successfully outputs 0 (i.e., $\pi_i \leq \Delta$). We observe that the method for bias correction makes significant improvements for estimating PageRank. We computed the fundamental matrix for the PageRank MC and observed that that $Z_{\max}(i) \approx 1$ for all $i$.

Figures 2(c) and 2(d) show the computation time, or total number of random walk steps taken by our algorithm, as a function of $\Delta$. Each figure shows the results from three different nodes, chosen to illustrate the behavior on nodes with varying $\pi_i$. The figures are shown on a log-log scale. The results confirm that the computation time of the algorithm is upper bounded by $O(\frac{1}{\Delta})$, which is linear when plotted in log-log scale. When $\Delta > \pi_i$, the computation time behaves as $\frac{1}{\Delta}$. When $\Delta < \pi_i$, the computation time grows slower than $O(\frac{1}{\Delta})$, and is close to constant with respect to $\Delta$.

## 5  Summary

We proposed a local algorithm for estimating the stationary probability of a node in a MC. The algorithm is a *truncated Monte Carlo* method, sampling return paths to the node of interest. The algorithm has many practical benefits. First, it can be implemented easily in a distributed and paral-lelized fashion, as it only involves sampling random walks using neighbor queries. Second, it only uses a constant size neighborhood around the node of interest, upper bounded by $\frac{1}{\epsilon\Delta}$. Third, it only performs computation at the node of interest. The computation only involves counting and taking an average, thus it is simple and memory efficient. We guarantee that the estimate $\hat{\pi}_i^{(t)}$, is an upper bound for $\pi_i$ with high probability. For MCs that mix well, the estimate will be tight with high probability for nodes such that $\pi_i > \Delta$. The computation time of the algorithm is upper bounded by parameters of the algorithm, and constant with respect to the size of the state space. Therefore, this algorithm is suitable for MCs with large state spaces.

**Acknowledgements:** This work is supported in parts by ARO under MURI awards 58153-MA-MUR and W911NF-11-1-0036, and grant 56549-NS, and by NSF under grant CIF 1217043 and a Graduate Fellowship.

## Footnotes

[1]Throughout the paper, *Markov chain* and *random walk* on a network are used interchangeably; similarly, *nodes* and *states* are used interchangeably.

[2]We use the notation $\tilde{O}(f(a)g(b))$ to mean $\tilde{O}(f(a))\tilde{O}(g(b)) = \tilde{O}(f(a)\text{polylog}f(a))\tilde{O}(g(b)\text{polylog}g(b))$.

[3]The bound for $t_{\max}$ is always true (stronger than with high probability).

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
