[Reviews · NeurIPS 2013]

Submitted by Assigned_Reviewer_4

This paper describes a novel procedure to test if the stationary probability of a chosen state in a Markov Chain is larger or smaller than some specified value. The procedure operates by taking a collection of length-bounded random walks starting at the chosen state, and geometrically increasing the length until some stopping condition is satisfied.

The paper is well structured with a good description of the problem statement and a good motivating example of the difficulty of the problem. The algorithm is well described and easy to follow, with some small experiments that demonstrate the empirical correctness of the procedure.

However, while theoretically interesting, I am finding it difficult to come up with a setting in which this could be useful in practice. For small simple distributions such as that evaluated in the examples, the complete stationary distributions can either computed exactly, or approximated using MCMC extremely efficiently. For larger, more interesting and more complex distributions, the stationary probability of any given state will be so small as to make the algorithm excessively costly. Furthermore, I am not sure what is the setting in which I am already provided with a "guess" of the stationary probability of a given state which I need to test.

Perhaps I can use this to do a "binary search" to estimate the stationary probability of some arbitrary state of a distribution, thus estimate (or bound) the partition function?


Other issues:
- The "bias correction" is not well described or justified. The first paragraphs of page 6 "Justifying the stopping condition and bias correction" does not provide enough intuition.
- The implications of Assumption 3.4 is not particularly obvious.

Minor issues:
- Page 4: line 163: "choosing an \epsilon, k" might be better phrased as "fixing an \epsilon, k", otherwise it seems like you are allowing \epsilon and k to vary.
- Page 5: line 261 P_{ii} is defined on page 3, but P_{ii}^{t} is not really defined?
Summary: Overall well written and theoretically interesting, but not particularly practical.

Submitted by Assigned_Reviewer_6

This paper introduces a new randomized algorithm for deciding properties of the stationary distribution of discrete time, irreducibile, aperiodic, positive recurrent Markov Chains (MC). Specifically, it considers the problem of deciding whether a state of a MC has a probability (according to the stationary distribution) that is larger than some threshold. The idea is to estimate this probability by running a sequence of random walks from the state, truncated at geometrically increasing length thresholds. The length of these random walks is used to estimate the return time, which is then used to estimate the stationary probability. Some correctness and convergence properties of the algorithm are proved by the authors, and are evaluated in simulation for a PageRank and an MM1 queue MC.

This is mainly a theoretical paper. Although the basic idea is fairly simple and intuitive (which is a good thing), the proofs of correctness and convergence provided by the authors are based on a non-trivial theoretical analysis of the concentration of return times both for finite and countable states Markov chains. The theoretical analysis is the main contribution of the paper.

The paper is generally well written and easy to read. The authors provide examples and the intuition behind their theoretical results, which facilitates the reading.
The problem considered is important and relevant for NIPS audience. However, I have some concerns about the practical relevance of the results.

In theorem 3.1, (1) rules out false negatives with high probability. It seems that it leaves open the possibility of false positives, i.e. a state with stationary probability smaller than Delta on which the algorithm outputs 1. Later line 286, it says “for low probability nodes, the algorithm will terminate at stopping condition (a)“ which seems to rule out false positives. The correctness of the algorithm (or the conditions required for that) is an important point that should be clarified, as it significantly affects its usefulness in practice. Related to this, the limitation counterexample (end of section 2.2) is interesting. It would be good to discuss more in detail how it relates to your theoretical results.

If the estimation is accurate only for MC that mix well, what is the an advantage over running MCMC to estimate the stationary distribution? Related to this, it seems to me that in many applications, one wants to find high probability states, rather than deciding if a given state has high probability or not. Would your algorithm still be useful in this case?

The experimental evaluation is a bit disappointing. The simulation results provide empirical evidence for the theoretical results, but the new technique is not really compared to other methods (e.g., regular MCMC, power method for finite states or specialized PageRank techniques mentioned in the introduction). It would have been good to show that it can provide answers on certain problems of interest where traditional techniques fail. For example, using it for a probabilistic inference task for some discrete probabilistic model with an exponentially large number of states, which is a typical sampling application.

It is known that sampling is generally hard and believed to be intractable in the worst case, and estimating a stationary probability would give information on the normalization constant (partition function), which is also intractable to compute in the worst case (#P complete). I wonder what is the hardness of the decision problem considered here? I think it would be good to give a sense of where the theoretical results stand in terms of known complexity results.
Summary: A theoretical paper introducing a new truncated Monte Carlo method to estimate some properties of the stationary distribution of Markov Chains. It is unclear if the new method has practical advantages over traditional techniques.

Submitted by Assigned_Reviewer_7

The authors present an algorithm to estimate (with potentially arbitrary accuracy) whether the stationary distribution of a specific state of a Markov chain is greater than or less than a pre-specified parameter, \delta. The algorithm uses truncated Monte Carlo rollouts to estimate the expected first return time from the specified state and then inverts this to get the state's stationary distribution. The paper also provides a novel theoretical contribution in the form of a proof that return times have exponentially decaying tails for countable-space MCs (not just finite-space MCs, which was what was known previously). A minimal empirical evaluation is provided as well, to demonstrate that this algorithm can be run.

Overall, the paper is quite well-written and clear with good explanations and intuitions given behind specific choices or aspects of the work. The authors address potential questions in good fashion and try to give an honest presentation of their work. That being said, I have a number of questions regarding the work.

Regarding the abstract, the authors state that power iteration and MCMC run in linear time, but they also give the stationary distribution over all states, so if you did that with this algorithm it would be at least linear in the state space size as well, no?

Regarding practicality, could the author's give some intuition about what someone would use this procedure for as well as how to set the parameters for different types of use cases? The authors indicate that there is wide applicability of this algorithm, which is true, but I am having a difficult time coming up with useful purposes of this algorithm (unless one runs it on every state in the space, which is somewhat unreasonable). Could you comment on potential future extensions (potentially to improve practicality, e.g., if there was smoothness in the space)? Finally, the Borgs et al. [17] algorithm cited in the paper seems more practically relevant -- could this algorithm be extended in that direction somehow?

Why doesn't the algorithm return the estimated stationary probability of i, instead of just returning 0/1 based on \delta? It seems that that would be more useful.

Specifics:
- it would be helpful if the description of the algorithm was a bit clearer, specifically
- specify the inputs as the parameters and the state i
- clearly state that s_k is the length of a sampled trajectory

- the statement of theorem 3.1 is strangely asymmetrical. Why don't the authors just use "With probability greater than (1-\alpha): If \pi_i < \delta, the algorithm outputs 0 and if \pi_i > \delta the algorithm outputs 1."

- on line 261, P^(t)_{ii} is introduced without being defined (as far as I can find) -- P in all other instances is not time-dependent (and thus Z_ii should tend to +/- \infty, no?) -- what is this quantity?

- furthermore, on line 280, the authors state that, in PageRank, Z_ii \approxeq 1, but give no intuition about whether this is good/bad or high/low -- what is the range of Z_ii?

- there is a duplicated sentence on life 653 of the supplementary material

Results section:
- change colors and data markers to be distinguishable in black and white
- does the x-axis of (a) refer to the # of nodes or the node IDs?
- is bias correction just working better because it is taking more steps (based on (b)) than the biased estimate? can you control for this?
- please show the actual # steps and \delta (not the log of them) but plot these values on a log scale (so the reader has a better understanding of the numbers being shown -- figure (b) seems to indicate that 8 million steps were taken for some of the data points?)
- a log scale on the y-axis of (c) would give a better indication of the differences between the data points
- the figures are numbers from a single run on a single graph? why aren't they averaged over many random graphs?
- a real data example would help highlight the efficacy of this algorithm in domains with strange distributions and hot spots in the state space
- (b) and (d) are said to show the results from three different nodes, but what are the results for each and how are they combined for the graph? could all three be shown separately?
- it's not clear that the computation time is upper-bounded by O(1 / \delta) because it's a log-log scale -- could the authors draw this line on the graph?
Summary: The paper presents a truncated Monte Carlo method for estimating the stationary distribution of a specific state of a Markov chain by exploiting the inverse relationship between a state's stationary distribution and its expected first return time. This is a clear, well-written paper that provides an interesting theoretical contribution but is less convincing as a practical algorithm.
Author Feedback

Author rebuttal: Thank you so much for your thoughtful and helpful comments. We appreciate and will incorporate your suggestions for improving the clarity of the results and impact of the simulations.

1) Output

The algorithm does output an estimate for the stationary probability of i, given by \hat{\pi}_i at the last iteration t_max. The accuracy of this estimate is well understood and characterized in Thm 3.2 and 3.5. We emphasize the 0/1 output depending on \Delta to highlight the role of the stopping conditions on the estimates. Since stopping condition (a) causes the algorithm to terminate once \hat{\pi}_i < \Delta/(1+\epsilon), for nodes such that the estimate is less than \Delta/(1 + \epsilon), we can only conclude that the true \pi_i is also less than \Delta.

2) Practical Applications or Settings

Our method is similar to other MCMC type methods. The key advantage of our algorithm is that it is “local”, i.e. uses only a local neighborhood of a node i. The only action performed on the data involves neighbor queries corresponding to each random walk step. In addition, our algorithm is “node centric”, it only needs computations to be performed at the node of interest. This allows us to estimate stationary distributions of extremely large graphs without loading the entire graph into memory. The accuracy of the estimate depends on the radius of the local neighborhood and graph properties according to Thm 3.2. Our algorithm also does not need to know the size of the state space, n.

One may only have access to a local neighborhood within a network due to constraints for accessing or storing the information. In settings such as the web graph or citation networks, and building the network structure first involves crawling the entire network and storing a representation of it. In social network platforms such as Facebook or LinkedIn, there may be privacy constraints that prevent a third party user to access the global network. However, it may be possible to obtain access to local neighborhoods by gaining permission from individuals or by crawling publicly accessible profiles. Thus it is useful to have an algorithm with clear theoretical guarantees for estimating stationary probabilities of nodes within the local neighborhood given the strict access constraints. There are settings where a user may be interested in computations of a smaller scale for a subset of the nodes within the context of a large network. If a local business owner wants to compute his approximate Pagerank, he can focus on his local network rather than build a network for the entire internet.

3) Limitations and correctness of our Method

Our method is a Monte Carlo type method over a positive recurrent Markov chain. Any Monte Carlo sampling technique will perform poorly on a Markov chain that mixes slowly, since the random walk will get stuck within local substructures of the network. Our algorithm guarantees with high probability that the output estimate is an upper bound, whereas the results of standard MCMC methods are very sensitive to the initial distribution. Despite the restriction of our algorithm to a local neighborhood, our algorithm does not do much worse than standard MCMC methods, and the error arising from the truncation is characterized in Thm 3.2. Previous MCMC methods have not been analyzed on countably infinite state space Markov chains.

4) When does the algorithm produce “false positives”?

There are two stopping conditions that determine the 0/1 output of the algorithm. The algorithm terminates at stopping condition (a) when the estimate drops below \Delta. The algorithm terminates at stopping condition (b) when P(T_i > \theta) * \hat{\pi}_i < \epsilon \Delta. Stopping condition (b) is always satisfied for \theta > \frac{1}{\epsilon \Delta}, which leads to Thm 3.1(2). From the example in Figure 1a, we see that it is possible for n >> \frac{1}{\epsilon \Delta} such that it is impossible to determine whether \pi_i is greater or less than \Delta while restricted to the local neighborhood. The algorithm will terminate at stopping condition (b) before stopping condition (a) is satisfied, and will thus output 1, though \pi_i may be less than \Delta. This is unavoidable for local algorithms.

5) Extensions to improve practicality

The algorithm has a natural extension of counting frequencies of visits to other nodes along the sample paths to obtain estimates for nodes other than the starting node. This extension can be used to find high probability states. This has been excluded due to space constraints.

6) Intuition about the Bias Correction

The bias corrected estimate does not take any more steps. It multiplies the original estimate by the fraction of samples that indeed returned to node i. This is based on the property that \pi_j is equal to the expected number of visits to j along a sample path that begins at node i and returns to node i, divided by E_i[T_i], for any nodes i and j. Thm 3.2 states that the original estimate tends to overestimate \pi_i by P(T_i > \theta) * Z_ii * \hat{\pi}_i. Therefore, after subtracting P(T_i > \theta) *\hat{\pi}_i for the bias correction, the new estimate will have a reduced error of P(T_i > \theta) * (Z_ii - 1) * \hat{\pi}_i.

7) Parameters

With probability 1 - \alpha, the sample average will be within a 1 \pm \epsilon interval around its mean. Set parameters to the desired confidence and accuracy within the computational budget. This does not change between use cases. Setting \Delta to 0.01 means that the algorithm does not distinguish among nodes that have less than 1% of the stationary probability “weight” over the state space.

8) Complexity Results
A wide body of research is dedicated to this, see “The Markov chain Monte Carlo method: an approach to approximate counting and integration, Mark Jerrum and Alistair Sinclair, 1996”. MCMC algorithms provide approximation guarantees for such hard counting problems.